# The Molecular Comorbidity Network of Periodontal Disease

**DOI:** 10.3390/ijms251810161

**Published:** 2024-09-21

**Authors:** Mireya Martínez-García, Enrique Hernández-Lemus

**Affiliations:** 1Department of Immunology, National Institute of Cardiology ‘Ignacio Chávez’, Mexico City 14080, Mexico; mireya.martinez@cardiologia.org.mx; 2Computational Genomics Division, National Institute of Genomic Medicine, Mexico City 14610, Mexico; 3Center for Complexity Sciences, Universidad Nacional Autónoma de México, Mexico City 04510, Mexico

**Keywords:** periodontal disease, comorbidities, molecular mechanisms of systemic diseases, genetic associations, biological databases, inflammatory pathways

## Abstract

Periodontal disease, a multifactorial inflammatory condition affecting the supporting structures of the teeth, has been increasingly recognized for its association with various systemic diseases. Understanding the molecular comorbidities of periodontal disease is crucial for elucidating shared pathogenic mechanisms and potential therapeutic targets. In this study, we conducted comprehensive literature and biological database mining by utilizing DisGeNET2R for extracting gene–disease associations, Romin for integrating and modeling molecular interaction networks, and Rentrez R libraries for accessing and retrieving relevant information from NCBI databases. This integrative bioinformatics approach enabled us to systematically identify diseases sharing associated genes, proteins, or molecular pathways with periodontitis. Our analysis revealed significant molecular overlaps between periodontal disease and several systemic conditions, including cardiovascular diseases, diabetes mellitus, rheumatoid arthritis, and inflammatory bowel diseases. Shared molecular mechanisms implicated in the pathogenesis of these diseases and periodontitis encompassed dysregulation of inflammatory mediators, immune response pathways, oxidative stress pathways, and alterations in the extracellular matrix. Furthermore, network analysis unveiled the key hub genes and proteins (such as TNF, IL6, PTGS2, IL10, NOS3, IL1B, VEGFA, BCL2, STAT3, LEP and TP53) that play pivotal roles in the crosstalk between periodontal disease and its comorbidities, offering potential targets for therapeutic intervention. Insights gained from this integrative approach shed light on the intricate interplay between periodontal health and systemic well-being, emphasizing the importance of interdisciplinary collaboration in developing personalized treatment strategies for patients with periodontal disease and associated comorbidities.

## 1. Introduction

Periodontal disease, a chronic inflammatory condition affecting the supporting structures of the teeth, has long been recognized as a significant global public health concern. Beyond its impact on oral health, accumulating evidence suggests that periodontal disease is intricately linked to various systemic conditions, thereby emphasizing the need for a deeper understanding of its molecular comorbidities [1]. The recognition of the shared pathogenic mechanisms between periodontal disease and systemic diseases holds profound implications for both clinical management and therapeutic interventions.

Periodontal disease is an extremely prevalent condition among adults [2,3], yet is not uncommon in children [4,5]. There is a diverse spectrum of inflammatory ailments that affect the periodontium, an overarching term encompassing gingiva, cementum, periodontal ligament, and alveolar bone [6]. Periodontitis typically commences with an uncontrolled inflammatory response triggered by gradual bacterial colonization of tooth surfaces and soft gingival tissues—referred to as Gingivitis [7]. However, it is generally thought that it is the host’s inflammatory reaction to microbial challenges that primarily instigates the degradation of the periodontium, leading to periodontitis [8].

In periodontitis, pathogens incite leukocytes of the innate immune system to release pro-inflammatory mediators like cytokines, which significantly contribute to the progression of chronic periodontitis [9]. Although infection is a prerequisite for periodontitis, it alone does not suffice for disease advancement. Pathogens instigate the acquired immune system, exacerbating the inflammatory condition’s progression and inflicting severe damage on soft and hard periodontal tissues [9]. Individual susceptibility tied solely to organismal immune and inflammatory responses is deemed crucial, as most signaling pathways and cellular events common to these disorders can be traced back to concurrent molecular origins [6,10].

### 1.1. The Genetic and Molecular Origins of Periodontitis

Genetic factors linked to susceptibility, severity, and progression of periodontitis have identified various gene variants of interest [11,12]. Notably, certain genetic polymorphisms in genes like Interleukin-1 beta (IL1B), interleukin 1 receptor antagonist (IL1RN), the Fc gamma receptor III subunit b (FcγRIIIb), vitamin D receptor (VDR), and Toll-like receptor 4 (TLR4) have been associated with aggressive periodontitis susceptibility, while others in IL1B, IL1RN, interleukin 6 (IL6), interleukin 10 (IL10), VDR, the cluster of differentiation 14 glycoprotein (CD14), TLR4, and and the matrix metalloprotease 1 (MMP-1) are linked to chronic periodontitis [13]. The heritability of periodontal disease susceptibility has been discussed extensively, with studies estimating a heritability of up to 50% after adjusting for behavioral and environmental factors, although no evidence of heritability has been found in gingivitis, a precursor to periodontitis [14]. Genome-wide association studies have provided valuable insights into the genetics of periodontal disease, revealing heritability dependencies and identifying associated loci, with most of them located in non-coding genomic regions that likely modulate gene expression through regulatory interactions [15]. However, pinpointing causal variants remains challenging in genome-wide association studies (GWAS), hindering elucidation of the molecular mechanisms crucial for diagnostics and therapeutics. Nevertheless, systematic reviews and meta-analyses have identified significant associations between genetic variants in genes like interleukin 1A (IL-1A), interleukin 1B (IL-1B), IL6, IL10, the matrix metalloprotease 3 (MMP-3), and matrix metalloprotease 9 (MMP-9) and periodontitis risk [16].

Furthermore, investigations combining periodontal disease with other conditions, such as cardiovascular diseases, have uncovered potential shared genetic backgrounds. Studies have identified genes like the antisense non-coding RNA in the INK4 locus (ANRIL/CDKN2B-AS1), plasminogen (PLG), and the Calmodulin-binding transcription activator 1 (CAMTA1)/Vesicle-Associated Membrane Protein 3 (VAMP3) implicated in both periodontitis and coronary artery disease pathogenesis, shedding light on functional features and accounting for some missing heritability of periodontal disease [17]. Additionally, joint genetic and functional studies have highlighted disrupted immunogenetic blueprints associated with inflammation, revealing a signature of over 65 genes involved in inflammatory processes and their association with cardiovascular diseases [18]. Genes related to bone morphogenetic and developmental processes have also been implicated in periodontal disease. For instance, single nucleotide polymorphisms (SNPs) in the Bone Morphogenetic Protein 2 (BMP2), Bone Morphogenetic Protein 4 (BMP4), Mothers against decapentaplegic homolog 6 (SMAD6), and Runt-related transcription factor 2 (RUNX2) genes have been significantly associated with persistent apical periodontitis, suggesting epistatic interactions contributing to increased risk [19]. Moreover, larger genetic variants like long runs of homozygosity (LROH) spanning multiple genes have been linked to various stages of periodontitis severity [20].

The complex relationship between microbial infections and hosts, particularly in chronic infections like those underlying periodontal disease, involves intricate molecular mechanisms triggering and sustaining infection-associated inflammation [21]. Genetic immunodeficiencies like leukocyte adhesion deficiency-1 (LAD1) result in severe periodontal inflammation and bone loss, with recent evidence implicating enhanced inflammatory responses mediated by interleukin 17 (IL17) [22,23]. Genetic variants in a chemokine receptor Interleukin 8 receptor, beta (CXCR2) have also been associated with susceptibility to prolonged periodontal bacteremia leading to chronic periodontitis [24].

### 1.2. The New Classification of Periodontal Disease Provides a Clearer Context for Comorbidity Conditions

The new 2018 classification of periodontal disease, developed by the American Academy of Periodontology (AAP) and the European Federation of Periodontology (EFP), represents a significant shift from previous schemes by integrating both clinical features and risk factors, including systemic conditions that influence periodontal disease progression and treatment outcomes [25]. In the EFP/AAP case definition system, the relevance of molecular comorbidity is evident in its acknowledgment of periodontitis as not only a localized oral disease but also as a condition with significant systemic implications. In this context, the EFP/AAP classification emphasizes the importance of a multidisciplinary approach that considers both oral and systemic health, reflecting a more integrated view of patient care. The EFP/AAP case definition system has been shown to have a greater ability to detect well-established forms of periodontitis related to the presence and severity of various systemic diseases such as cardiovascular disease, diabetes, and chronic obstructive pulmonary disease [25,26].

Using a unanimously accepted case definition of periodontal status, such as the new 2018 AAP/EFP classification system, is crucial for future studies exploring the relationship between periodontitis and systemic diseases. Furthermore, at the molecular level, the new classification system considers factors like the role of inflammation, immune response, and genetic susceptibility, which are critical when studying the shared molecular pathways between periodontitis and systemic diseases. A standardized definition enables researchers to correlate molecular biomarkers, genetic variants, or inflammatory mediators consistently across studies. This uniformity is especially important for identifying shared molecular mechanisms, such as cytokine networks or oxidative stress pathways, which may link periodontitis with conditions like cardiovascular diseases, diabetes, or rheumatoid arthritis. Such uniformity is essential to ensure that studies across different populations, research settings, and geographical locations are comparable and interpretable in a meaningful way [25,26].

### 1.3. Periodontitis and Systemic Health

Historically, the association between oral health and systemic health has been debated, with early observations often being dismissed as anecdotal. However, over the past few decades, epidemiological studies have consistently demonstrated compelling associations between periodontal disease and a spectrum of systemic conditions, including cardiovascular diseases, diabetes mellitus, rheumatoid arthritis, and inflammatory bowel diseases. These associations transcend mere coincidence and suggest underlying molecular connections that warrant exploration.

The advent of high-throughput technologies and bioinformatics tools has revolutionized our ability to dissect complex molecular networks underlying disease pathogenesis. In this context, integrative approaches that leverage literature mining and biological databases provide valuable insights into the shared molecular signatures and pathways between periodontal disease and its comorbidities. By systematically analyzing genetic associations, protein interactions, and pathway crosstalk, researchers can unravel the molecular underpinnings of these complex relationships.

Periodontal disease is also a widespread and significant public health concern, affecting a substantial portion of the global population. It is estimated that moderate to severe periodontitis impacts nearly 50% of adults over the age of 30, with prevalence increasing with age. This chronic inflammatory condition, which leads to the destruction of the supporting structures of the teeth, is not only a leading cause of tooth loss but also has far-reaching implications for systemic health. The inflammatory nature of periodontal disease means that its effects are not confined to the oral cavity; instead, the chronic inflammation and immune responses associated with periodontitis can have systemic consequences, contributing to the development and exacerbation of various chronic conditions.

The systemic impact of periodontal disease is particularly concerning given its association with several major health issues, including cardiovascular disease, diabetes mellitus, rheumatoid arthritis, and adverse pregnancy outcomes. Research has shown that the chronic inflammation seen in periodontitis can exacerbate systemic inflammatory responses, leading to worsened outcomes in these conditions. For example, individuals with periodontitis are at a higher risk for heart disease and stroke, partly due to the systemic dissemination of inflammatory mediators and pathogens from the oral cavity. Similarly, the bidirectional relationship between periodontal disease and diabetes highlights the complex interplay between oral health and systemic metabolic control. Understanding the molecular comorbidities of periodontal disease is therefore crucial, not only for preventing oral health deterioration, but also for mitigating its broader impact on overall health. This underscores the importance of research that delves into the shared molecular mechanisms between periodontitis and its associated systemic conditions, paving the way for more effective, holistic approaches to healthcare that integrate the management of oral and systemic diseases.

Given these issues, analyzing the comorbidity spectrum around periodontal diseases from an integrative perspective has become a relevant endeavor. Hajishengallis and Chavakis have recently [27] summarized a great deal of work in connection with inflammation-related periodontitis comorbidities. Peridontal disease comorbidities have been also related with inflammation based in immune dysfunction and clonal hematopoiesis [28]. Teles and coworkers, in turn, have explored how human viruses can be the link between periodontitis and a number of diseases [29]. The role of shared morbidity mechanisms has also been discussed in connection with novel therapeutic interventions [30].

This paper aims to review and synthesize the current knowledge on the molecular comorbidities of periodontal disease, with a particular focus on diseases sharing associated genes, proteins, or molecular mechanisms. By integrating data from diverse sources, including DisGeNET2R, Romin, and Rentrez R libraries, we seek to elucidate the intricate interplay between periodontal health and systemic well-being at the molecular level. Furthermore, we aim to identify potential therapeutic targets and pathways for intervention, thereby paving the way for personalized treatment strategies tailored to individuals with periodontal disease and associated systemic conditions.

Given the growing recognition of the systemic implications of periodontal disease, this study thus aims to systematically identify and analyze the molecular comorbidities associated with periodontitis. Specifically, our research seeks to identify diseases that share common genetic, protein, or molecular pathway associations with periodontal disease through comprehensive literature and database mining. Additionally, we aim to elucidate the shared pathogenic mechanisms underlying periodontitis and its associated comorbidities, with a focus on the roles of biological actors such as inflammatory mediators, immune response pathways, oxidative stress, and extracellular matrix alterations. Furthermore, we intend to map the molecular comorbidity network of periodontal disease to uncover key hub genes and proteins that may serve as potential therapeutic targets. By constructing a detailed diseasome network, we also aim to provide insights into the systemic nature of periodontal disease, illustrating the interconnectedness of periodontitis with various systemic conditions. Through these objectives, this study seeks to deepen the understanding of the molecular interplay between periodontal health and systemic diseases, paving the way for the development of personalized treatment strategies that consider both oral and overall health.

## 2. Results

### 2.1. Molecular Comorbidity Network

The molecular comorbidity network of periodontitis encompasses 2219 curated gene–disease interactions involving 80 different disease conditions. The top 10 conditions that are comorbid with periodontitis, ranked by the probability of their leading gene interactions being loss-of-function intolerant (PLI), include the following: malignant neoplasm of breast, breast carcinoma, malignant neoplasm of prostate, atrial fibrillation, depressive disorder, obesity, adenocarcinoma, diabetes mellitus, non-insulin-dependent diabetes, rheumatoid arthritis, and Alzheimer’s disease. The PLI score, derived from large-scale sequencing data like the Exome Aggregation Consortium (ExAC), reflects the likelihood that a gene is intolerant to loss-of-function mutations, which could result in severe phenotypic effects. High PLI scores indicate potential relevance to disease susceptibility or pathogenesis, and such genes should be prioritized for further investigation. For the full annotated list of interactions, refer to Appendix A.

### 2.2. Periodontitis-Centered Disease Network

Figure 1 presents a periodontitis-centered comorbidity network, which underscores the systemic nature of periodontal disease by highlighting its diverse comorbidities. These range from oral conditions like Pericementitis to inflammatory, immune, cardiovascular, neoplastic, and psychiatric conditions, including substance abuse. The gene-sharing plot (Figure 2) shows that some diseases share up to 25% of their associated genes with periodontitis, reinforcing the extensive overlap in their molecular underpinnings.

### 2.3. A Diseasome of Periodontitis-Related Conditions

The periodontitis-associated diseasome, visualized in Figure 3, provides a comprehensive view of the interconnected genetic and molecular factors among comorbid diseases. Most diseases and genes form a highly interconnected “core”, with few remaining on the “periphery”. The top diseases associated with a higher number of shared genes with periodontitis include malignant neoplasm of breast (278 genes), breast carcinoma (260), malignant neoplasm of prostate (164), diabetes mellitus, non-insulin-dependent (116), and others. Key genes like the tumor necrosis factor (TNF), interleukin 6 (IL6), and prostaglandin-endoperoxide synthase 2 (PTGS2) are implicated in multiple diseases, suggesting their central role in the molecular diseasome network (Appendix A).

Diseases such as Alzheimer’s disease, depressive disorder, and breast carcinoma have a high degree of gene overlap with periodontitis, indicating closer molecular associations. In contrast, conditions like acute coronary syndrome and adrenal gland hypofunction show limited gene overlap, though even minimal shared genes can indicate significant molecular connections. The distribution of shared genes highlights the complexity of disease interactions (Figure 4) and underscores the multifactorial nature of both periodontitis and systemic diseases.

The top 10 genes (actually 11 genes because of ties, see Table 1) implicated in the higher number of diseases (also in parentheses) in this diseasome are the following: *TNF* (24), *IL6* (23), *PTGS2* (14), interleukin 10 *IL10* (13), nitric oxide synthase 3 *NOS3* (12), interleukin 1-beta *IL1B* (12), vascular endothelial growth factor A *VEGFA* (11), the B-cell lymphoma 2 protein *BCL2* (11), the signal transducer and activator of transcription 3 *STAT3* (11), leptin *LEP* (11), and tumor protein P53 *TP53* (11). For a comprehensive list please refer to Appendix A.

Some diseases exhibit a higher number of shared genes with periodontitis compared to others. For example, diseases like Alzheimer’s disease, depressive disorder, and breast carcinoma show a relatively large number of shared genes with periodontitis, suggesting closer molecular associations or shared genetic underpinnings (Appendix A).

On the other hand, certain diseases have a smaller number of shared genes with periodontitis. Conditions like acute coronary syndrome, acute monocytic leukemia, and adrenal gland hypofunction exhibit a limited overlap in genes with periodontitis within the diseasome network. However, even diseases with fewer shared genes may still have significant implications for understanding disease pathogenesis and comorbidities, as even a small number of shared genes can indicate potential molecular connections or common biological pathways (see Appendix A).

Overall, the distribution of shared genes across different diseases in the diseasome network underscores the complexity of disease interactions and highlights the multifactorial nature of both periodontitis and systemic diseases. Such is the case, for instance, of BCL2 (involved in such disparate conditions within the periodontitis diseasome such as Alzheimer’s disease, polycystic ovary syndrome, myocardial infarction and several tumors such as adenocarcinoma, malignant neoplasm of breast, multiple myeloma, malignant Head and neck neoplasm, esophageal neoplasms, and malignant neoplasm of prostate); of the estrogen receptor 1 (ESR1) (involved in myocardial infarction, depressive disorder, breast carcinoma and atherosclerosis, to name a few); and of the G glutathione S-transferase Mu 1 (GSTM1) (associated with Parkinson’s disease, leukemia, diabetes mellitus, non-insulin-dependent, and asthma). These are just come examples representative of the diversity of conditions covered in the periodontitis diseasome network (further information can be found in Appendix A).

The varying degrees of overlap in genes between periodontitis and different diseases suggest heterogeneous molecular relationships and underscore the importance of considering diverse factors, such as genetic predisposition, environmental influences, and shared biological processes, in understanding disease comorbidities and developing targeted therapeutic interventions.

Looking at groups of somewhat similar diseases in the list provided in Appendix A, we can observe several clusters based on disease categories or shared pathophysiological mechanisms. In order to provide a systematic view of our findings, we will present them grouped in the following general categories:

Several disease clusters can be observed based on shared pathophysiological mechanisms:

**Cardiometabolic Diseases:** Conditions like acute coronary syndrome, atrial fibrillation, and coronary artery disease share genes related to inflammation, endothelial dysfunction, or immune dysregulation, common to both periodontal and cardiovascular disorders. Other relevant conditions include diabetes mellitus, non-insulin-dependent, non-alcoholic fatty liver disease, and dyslipidemias.

**Inflammatory and Immunological Diseases:** Diseases such as rheumatoid arthritis, Crohn’s disease, and asthma share inflammatory pathways with periodontitis, indicating potential immune-mediated processes affecting both oral and systemic health.

**Cancer:** Types of cancer, including breast cancer, cervical cancer, and adenocarcinoma, share genes with periodontitis, suggesting common pathways related to cell cycle regulation, tumor progression, or immune evasion.

**Neurological and Psychiatric Disorders:** Disorders like Alzheimer’s disease and depressive disorder suggest potential bidirectional relationships with periodontitis, mediated by systemic and neuroinflammatory responses.

**Other Diseases:** Conditions like pneumonia, osteoporosis, and chronic kidney diseases also share relevant molecular players with periodontitis.

These clusters reflect commonalities in pathophysiological mechanisms, such as inflammation, immune dysregulation, and cell proliferation, contributing to their co-association with periodontitis.

More specific information can be derived from the top genes of significance within this diseasome, which can be found in Figure 5. One can see a number of well-known disease-associated genes, ranging from DNASes like DNASE1, tyrosine kinases such as the protein tyrosine phosphatase non-receptor type 22 PTPN22, PIK3CA, AKT2, and oncogenes such as breast cancer-associated protein 1 (BRCA 1); tumor suppressors like the phosphatase and tensin homolog (PTEN) and the Ataxia-telangiectasia mutated gene (ATM), all commonly associated with neoplasms; nuclear receptors such as the peroxisome proliferator-activated receptor gamma (PPARG) associated with metabolic diseases; and extracellular matrix glycoproteins such as Fibrillin-1 (FBN1), often found in inflammatory and collagen-related diseases, etc.

Further systemic functional information can be found in the heatmap shown in Figure 6. There, one can see a summary of the different biological functions carried out by the set of genes/proteins within the periodontitis diseasome. A prevalence of enzymes as well as signaling and kinase proteins can be seen to be associated with most of the different diseases. Still relevant, though less generalized, is the presence of transporter, nucleic acid-binding, and enzyme modulator proteins.

Other functional categories, however, even if not present in most of the diseases, are prevalent (as a class) within a given one. Such is the case of transcription factors and cellular structure proteins in acute coronary syndrome or G-protein-coupled receptors in rheumatoid arthritis, for instance.

### 2.4. Functional Enrichment of the Diseasome Genes

Enrichment analysis provides insights into the shared molecular mechanisms within the periodontitis diseasome (see Figure 7). A significant number of enriched biological features, including 221 Gene Ontology (GO) molecular function categories, 2720 GO biological process categories, and multiple molecular pathways (e.g., KEGG, Reactome, Wikipathways), are associated with the periodontitis diseasome (representative examples can be found in Table 2; for the full list, please refer to Appendix A). Notable enrichments involve inflammatory pathways, immune signaling, transcriptional regulation, and extracellular matrix remodeling, which are critical in both periodontitis and its associated diseases.

Specific transcription factors (TFs) like SMAD3 and PAX4 and regulatory non-coding RNAs (miRNAs) are significantly enriched, indicating their role in the regulation of disease-associated genes within the diseasome. The complexity is further enhanced by the presence of enriched proteins and protein complexes, which indicate potential molecular targets for future therapeutic interventions.

Overall, the enriched molecular functions, pathways, and processes highlight the systemic nature of periodontitis and its interconnected comorbidities, providing a deeper understanding of the molecular landscape and potential therapeutic targets for this multifactorial inflammatory disease.

We can also notice how different regulatory non-coding transcripts, corresponding to 38 miRNAs, are associated with the core genes in the periodontitis diseasome. Each one of these regulatory transcripts is able to regulate a large number of gene targets (from a few dozen to close to a hundred) in the periodontitis diseasome (to examine the full list of miRNAs and their gene targets, please refer to Appendix A). An additional dimension of complexity is the one formed by active proteins (113) and protein complexes (24) reported as significantly enriched in the periodontitis diseasome. Furthermore, in addition to the 80 diseases directly related to periodontitis by sharing associated genes, there is a staggering set of 506 different health conditions reported as significantly enriched in the set of genes in the periodontitis diseasome (see Appendix A).

## 3. Discussion

In recent years, numerous researches and clinical investigations have documented the connections between periodontitis and various systemic inflammatory conditions, including rheumatoid arthritis, Type 2 diabetes mellitus, and atherosclerosis [3]. The  molecular and physiological underpinnings of these relationships have been explored in previous sections. Now, we delve into their implications for the onset and progression of a wide spectrum of diseases, encompassing cardiovascular disease, gastrointestinal and colorectal cancer, diabetes and insulin resistance, Alzheimer’s disease, respiratory tract infection, and adverse pregnancy outcomes, among others [2,31].

Understanding the association between periodontitis and other systemic inflammatory diseases gains significance due to the availability of therapeutic interventions, such as cytokine-based treatment strategies, with potential benefits for both periodontitis and systemic health [32]. Immune markers linking chronic periodontitis and diabetes, such as glycation dynamics and TNF-α, have been identified as reliable indicators of inflammation in gingival crevicular fluid and serum [33]. The glycemic status has been linked to periodontitis through mechanisms of systemic inflammation [34]. Moreover, interactions between local inflammation and innate immune responses have been observed in bone marrow stem cells co-cultured with macrophages obtained from diabetic periodontitis patients, further emphasizing the interplay between these conditions [35]. Conversely, evidence suggests that periodontal treatment can mitigate systemic inflammation in type 2 diabetes [36].

Recent experimental (mouse) models have demonstrated how periodontitis can induce systemic inflammation, exacerbating atherosclerosis through mechanisms driving endothelial-mesenchymal transitions [37,38]. Similar mechanisms have been discussed in the clinical care of human patients with lacunar infarct [39]. In the following sections, we will further analyze different disease categories that share molecular origins with periodontitis.

### 3.1. Cardiometabolic Diseases

Consistent epidemiological evidence underscores the association between periodontitis and heightened risk for cardiovascular diseases [32,40]. Given the inflammatory nature of vascular diseases like atherosclerosis, where immune reactions mediated by cytokines contribute to pathogenesis, the link with periodontitis is not unexpected [32]. Chronic infections in the vessel wall can incite a pro-inflammatory milieu, driving autoimmunity to vascular cells and initiating atherosclerosis [41]. Consequently, the sustained inflammatory state of periodontitis correlates with increased cardiovascular risk, with bacteremia and systemic inflammation precipitating endothelial lesions and vascular wall inflammation [39]. Moreover, periodontal treatment has been shown to reduce systemic inflammation and endothelial dysfunction, albeit with limited evidence of modifying atherosclerotic vascular disease outcomes [32,36].

Persistent periodontal inflammation exacerbates endothelial dysfunction and vascular inflammation [10,42]. Notably, inflammatory mediators like TNF-α and IL-6 impair nitric oxide production and endothelial function, contributing to atherosclerosis and CVD development [10]. Periodontitis is associated with inflammation; the molecular players common to both types of disease may indeed be related to pathways impacting endothelial function and vascular structure [10]. Severe periodontitis heightens risks for acute myocardial infarction and stroke, with periodontal treatment also significantly reducing their incidence [2,10].

Endocrine and metabolic dysfunction is another molecular and physiological component of the association between periodontitis and cardiometabolic diseases. Non-insulin-dependent diabetes mellitus, for instance, has been associated with periodontal disease for at least two decades [43,44]. This relationship indeed has many components, including the role of inflammation and signaling crosstalk [45,46], immune responses and immune cell infiltrates [47,48], and the effects of the microbiome [49,50,51], as well as epidemiologic and common risk factors [52,53,54]. Insulin resistance has also been considered in these associations [55,56,57]. Obesity is a complex trait that has been related to periodontitis [58]. Similarly, molecular, epidemiological, and lifestyle factors confirm this association [59,60,61,62].

Several conditions related to metabolic disruption have been discussed regarding their connections with periodontal disease; such is the case of dyslipidemias [63,64,65,66], and non-alcoholic fatty liver disease (now renamed Metabolic Dysfunction-Associated Steatotic Liver Disease (MASLD)) [67,68,69,70]. In the latter case, a clearer genomic molecular picture of the associated mechanisms has even been proposed [71]. Heart conditions with molecular links to periodontitis also include atrial fibrillation [72,73,74] and coronary heart disease [75,76,77,78,79], as well as vascular maladies such as essential hypertension [80,81,82,83,84,85] and peripheral arterial disease [86,87,88,89,90].

### 3.2. Inflammatory and Immunological Disorders

The comorbidity profile of periodontal disease encompasses various conditions, including autoimmune disorders like rheumatoid arthritis, systemic lupus erythematosus, and inflammatory bowel diseases. Disturbances in the delicate balance between innate and adaptive immunity, often triggered by chronic infections, may lead to autoimmune responses. For instance, in rheumatoid arthritis, patients exhibit increased bacterial loads and dysbiotic oral microbiota similar to those seen in periodontitis, with shared genetic susceptibility factors contributing to both conditions [3,91,92]. The inflammation observed in rheumatoid arthritis synovitis mirrors the immune response seen in periodontitis, involving a complex network of cytokines and immune cells that contribute to joint damage [93,94]. Additionally, periodontal pathogens like P. gingivalis may exacerbate rheumatoid arthritis progression by increasing mucosal permeability and facilitating immune cell migration to the joints [95]. Similar parallels exist between periodontitis and systemic lupus erythemathosus, where disrupted microbiome composition and upregulated pro-inflammatory cytokines contribute to disease pathogenesis [10,96].

Inflammatory bowel diseases, such as ulcerative colitis and Crohn’s disease, share common inflammatory pathways with periodontitis, potentially arising from dysbiotic intestinal microbiota and genetic predisposition [97,98]. Furthermore, periodontal disease has been linked to hematological conditions like leukemia and thrombocytopenia, where oral manifestations and impaired immune function contribute to disease progression and oral complications [99,100,101,102].

Other conditions related to immune and inflammatory changes that share molecular basis with periodontitis include psoriasis [103,104,105,106], IGA glomerulonephritis—though in spite of the molecular association, most studies have centered around the microbiological component of this association [107,108], Graves disease [109,110,111], and Sjogren’s syndrome [112,113,114]. Although, in this latter case, some studies have found inconsistencies and ambiguities between the epidemiological and biomarker findings [115,116,117].

### 3.3. Cancer

In recent years, numerous research and clinical studies have unveiled connections between periodontitis and several systemic conditions, including cancer [118]. While the precise relationship remains debated, individuals with periodontitis exhibit an elevated risk of fatal cancer, particularly lung cancer, and other consistent correlations have been observed for oral and esophageal carcinomas [119,120,121]. For breast cancer, meta-analyses have confirmed significant associations with periodontitis, although the risk diminishes in patients with a history of periodontitis who underwent periodontal treatment [122]. Importantly, it has been stated that periodontitis leading to tooth loss is associated with increased cancer risks [123,124].

Systemic inflammation, a hallmark of periodontitis, is implicated in various phases of cancer development, from initiation to metastasis, reflecting the complex interplay between inflammation, immune responses, and genetic instability [32]. Notably, studies have shown positive correlations with pancreatic, head and neck, and lung cancers [2,118]. Periodontal pathogens, particularly P. gingivalis and F. nucleatum, have been implicated in oral cancer pathogenesis through mechanisms involving direct interaction with oral epithelial cells and modulation of the host immune response [125]. Furthermore, chronic inflammation may influence carcinogenesis and cancer mortality, as observed in breast cancer, where periodontal inflammation promotes the recruitment of metastatic breast cancer cells [126,127].

Other neoplasms sharing some genetic origins with periodontal disease are prostate cancer (in particular the hereditary forms) [128]; however, an additional association related to periodontal pathogens has also been discussed [128].

### 3.4. Neurological and Psychiatric Diseases

Periodontitis exhibits intricate connections with neuroinflammation and neurodegenerative diseases, notably Parkinson’s and Alzheimer’s disease. Studies have revealed a higher prevalence of periodontitis in patients with Parkinson’s disease, suggesting a bidirectional relationship between neurological conditions and periodontitis [2,10,129,130,131]. Evidence suggests that pro-inflammatory cytokines released from ulcerated periodontal pockets may weaken the blood–brain barrier, facilitating the entry of inflammatory mediators into the brain and triggering neurodegenerative cascades [10,132].

Activated glial cells in the brain produce inflammatory cytokines similar to those implicated in AD, exacerbating neuronal damage caused by β-amyloid plaques and τ aggregates [133]. The presence of periodontal pathogens, such as P. gingivalis and T. denticola, in peri-postmortem human brains with AD further supports the role of these pathogens in brain inflammation associated with neurodegenerative diseases [2]. Moreover, the association between depression and chronic periodontitis underscores the involvement of systemic inflammatory processes in psychiatric conditions, implicating neuroinflammation in depressive disorders [42,134,135]. Inflammatory dysregulation is also involved in the sensitization of nociceptive fibres, thus leading to pro-pain processes that may unleash hyperalgesia [136,137,138,139].

### 3.5. Respiratory Diseases

Some diseases of the respiratory tract also share molecular and other factors with periodontitis [140,141,142]; such as asthma [143,144,145,146], pneumonia [147,148,149], sleep apnea [150,151,152,153,154], and chronic obstructive airway disease (COAD/COPD) [155,156,157,158]. We can notice, however, that most studies in the current literature (with some exceptions such as in COAD/COPD) consider these associations to have epidemiological origins rather than relying on their molecular and mechanistic origins, which gains support in view of our results presented above.

### 3.6. Musculoskeletal and Connective Tissue Diseases

More comprehensive—yet still incomplete—molecular and mechanistic links have been established between periodontitis and musculoskeletal and connective tissue diseases such as osteoporosis [159,160,161,162,163,164], amyloidosis [165,166,167,168], and Marfan syndrome [169,170,171,172,173]. The common axis to these maladies seems to be crosstalk between inflammation and tissue development pathways [174,175], involving oxidative stress in some cases deregulation [176,177,178,179,180].

### 3.7. Hormonal Diseases

Common genetic molecular origins between periodontitis and conditions related to hormone disruption have also arisen in our analysis. For instance, polycystic ovary syndrome [181,182,183,184], adrenal gland hypofunction [185,186,187,188], and hereditary angioedema type III [189,190,191,192]. Interestingly it seems that a common link between many of the molecular pathways involved is via crosstalk with the complementary system [193,194,195,196,197].

### 3.8. Kidney Diseases

Oxidative stress also plays a role in the connections between periodontitis and chronic kidney disease [198,199,200], under some circumstances via mechanisms driven by intertwining with gluthatione [201], phosphatidylcholine [202], or even vitamin D [203] metabolism. However, other mechanisms and risk associations have been found [204,205,206,207].

### 3.9. Biomolecular Foundations

It is relevant to highlight that the comorbidity conditions associated with periodontal disease that we have been studying and discussing in the present work are fundamentally related to shared molecular origins. An independent (but also quite relevant) analysis should be made for those comorbidity conditions that arise from common environmental conditions, risk factors or social determinants of health.

In the present context however, it is relevant to analyze the types and functions of the common molecules involved (see Figure 6) and the biological processes and functions spanned by these molecules (Figure 7). Regarding the former, we can see that most of the molecules involved are enzymes, signaling proteins, or kinases, i.e., they are related to metabolic and immune rather than developmental processes. This insight is further reinforced by looking at the myriads of statistically significant biological processes and molecular pathways associated with such molecules. This information, aside from its own basic research value, is also important for the design of targeted therapy. While abnormal development processes are difficult to cope with, metabolism and signaling phenomena can be (under certain circumstances) modified either by pharmacological interventions or by lifestyle changes (or both).

### 3.10. Some Open-Ended Avenues for Future Inquiry

Interestingly, some diseases in the network are seemingly unrelated to periodontitis from a clinical standpoint, such as acute monocytic leukemia, cervical cancer, and alcoholic intoxication. However, the presence of shared genes between these diseases and periodontitis (See Appendix A) suggests possible molecular connections or shared pathogenic mechanisms that warrant further investigation. As previously mentioned, the immune dysregulation and chronic inflammation characteristic of periodontitis may intersect with pathways involved in cancer development or hematological disorders, contributing to their co-association in the diseasome network.

The top 10 genes (actually 11 genes because of ties) implicated in the higher number of diseases (number of hits in parentheses) in this diseasome are the following: TNF (24), IL6 (23), PTGS2 (14), IL10 (13), NOS3 (12), IL1B (12), VEGFA (11), BCL2 (11), STAT3 (11), LEP (11), and TP53 (11). For a comprehensive list, please refer to Appendix A. One can see that, unsurprisingly, most of these molecules are associated with immunity and signaling pathways.

The hub genes identified in the periodontitis comorbidity network, such as TNF, IL6, PTGS2, IL10, and NOS3, play pivotal roles in connecting periodontitis with various systemic diseases, as indicated by their high connectivity degrees. TNF and IL6, which have connectivity degrees of 24 and 23, respectively, are among the most prominent hub genes. TNF is heavily involved in inflammatory pathways and immune regulation, which are critical in diseases such as rheumatoid arthritis, ulcerative colitis, diabetes mellitus, and myocardial infarction. Similarly, IL6 is known for its role in inflammation and is associated with conditions like Crohn’s disease, adenocarcinoma, malignant neoplasms of the breast, and atherosclerosis. The involvement of these genes in both periodontitis and other systemic conditions underscores the shared molecular mechanisms of inflammation and immune dysregulation, suggesting that therapies targeting these genes may have dual benefits for treating periodontitis and its comorbid conditions.

Other hub genes, such as PTGS2, IL10, and NOS3, also illustrate the complex interplay between periodontal health and systemic diseases. PTGS2, known for encoding Cyclooxygenase-2 (COX-2), has a significant role in inflammatory responses and is linked to conditions like depressive disorder, esophageal neoplasms, cerebrovascular accidents, and coronary artery disease. IL10, an anti-inflammatory cytokine, is involved in autoimmune and inflammatory diseases, including lupus erythematosus, myocardial infarction, and rheumatoid arthritis. The nitric oxide synthase gene NOS3 is connected with diseases such as Alzheimer’s disease, diabetes mellitus, coronary artery disease, and malignant neoplasms of the breast, highlighting its role in vascular function and oxidative stress. These genes’ involvement in diverse conditions points to the potential systemic effects of periodontal inflammation and the need for a broader perspective on patient management that considers both oral and systemic health.

Furthermore, the presence of other notable hub genes like VEGFA, BCL2, STAT3, LEP, and TP53, which are connected with multiple diseases, emphasizes the systemic impact of molecular pathways implicated in periodontitis. For instance, VEGFA is associated with psoriasis, adenocarcinoma, and Alzheimer’s disease, suggesting a link between angiogenesis, inflammatory responses, and neurodegeneration. The BCL2 gene, known for its role in cell survival, is implicated in diseases like polycystic ovary syndrome, diabetes mellitus, and various cancers. The interconnection of these genes with both periodontitis and systemic diseases reinforces the concept that periodontitis is not merely a localized oral condition but part of a broader systemic network influenced by shared genetic and molecular mechanisms. Understanding these hub genes’ roles can inform more effective and comprehensive strategies for preventing and treating periodontitis and its related comorbidities.

The extent to which these genes are able to modulate relevant biological functions, from inflammation, hormone function, metabolism, development and differentiation processes, to nutrient intake and processing, cell cycle and DNA repair, further highlights their relevance in multimorbidity associated with periodontal disease. This in turn helps us to provide a molecular and physiological basis [27,28,30,38,208] for the growing body of epidemiological evidence on these associations [209,210,211,212,213]. Thus, systematic exploration of the biological functions of these molecules in the context of such (and other potential) comorbidities seems to be an endeavor worth pursuing.

To build on the findings of this study, future research should focus on the experimental validation of the identified molecular mechanisms and therapeutic targets. Specifically, in vitro and in vivo studies could be designed to investigate the functional roles of the key hub genes and proteins identified in the periodontitis comorbidity network. For instance, CRISPR-Cas9 gene editing or RNA interference could be employed to knock out or downregulate these genes in cell cultures or animal models, allowing researchers to observe the resultant effects on periodontal inflammation and associated systemic conditions, such as cardiovascular disease or diabetes. Additionally, investigating the impact of overexpressing these genes in relevant models could provide insights into their role in disease progression and their potential as therapeutic targets.

Moreover, future studies could explore the therapeutic potential of targeting shared molecular pathways, such as inflammatory mediators, immune response pathways, and oxidative stress pathways, in the treatment of periodontitis and its comorbidities. This could involve the development and testing of small-molecule inhibitors, monoclonal antibodies, or other biologics that modulate these pathways. For example, clinical trials could be designed to evaluate the efficacy of drugs that target specific inflammatory mediators in patients with periodontitis and a high risk of cardiovascular disease. Additionally, large-scale genetic and transcriptomic studies in diverse populations could further elucidate the genetic variants and expression patterns associated with these pathways, offering a broader understanding of their role in disease susceptibility and progression. By pursuing these research directions, scientists can not only validate the findings of this study but also advance the development of personalized therapeutic strategies that address the complex interplay between periodontal disease and systemic health.

### 3.11. Assumptions and Limitations of the Present Study

The results of this study reveal statistically significant associations between periodontitis and a wide range of diseases based on shared genes, offering new insights into the potential molecular links between oral and systemic health. However, it is important to note that these associations do not establish causality. While the shared genetic components suggest potential connections, further research is needed to determine whether periodontitis directly contributes to the development of these diseases or if the associations reflect common risk factors or underlying mechanisms. Understanding the causal relationships is crucial for developing targeted therapeutic strategies that could address both periodontitis and its associated comorbidities.

Moreover, the analysis in this study relies heavily on the existing literature and databases, which may introduce biases, particularly toward well-studied diseases and genes. This could result in an under-representation of less-studied conditions or molecular mechanisms, potentially skewing the diseasome network. While the study successfully highlights shared genes between periodontitis and other diseases, it does not delve into the specific functional roles of these genes in the context of each disease. A more detailed investigation into the molecular pathways and mechanisms involved is necessary to fully understand the connections and their clinical implications. Additionally, the study uncovers a large number of enriched biological features within the periodontitis-associated diseasome, including various Gene Ontology (GO) categories and molecular pathways. This underscores the complexity of periodontitis but also presents a challenge in identifying the most relevant associations. The broad enrichment across nearly all cellular locations points to systemic phenomena, but this breadth may obscure the identification of specific cellular processes or locations that are particularly critical in the context of periodontitis. Furthermore, this study introduces a number of technical terms and concepts that may be difficult for non-experts to grasp, suggesting the need for more context or explanation to help readers understand the significance of the findings. Ultimately, while the study establishes associations between various regulatory elements, proteins, and health conditions with the periodontitis diseasome, it stops short of proving causality. Determining whether these elements play a direct role in periodontitis or are merely correlated with the disease remains an essential step for future research.

## 4. Materials and Methods

In this study, several key resources and repositories were utilized for data extraction to identify the molecular comorbidities associated with periodontal disease. The National Center for Biotechnology Information (NCBI) of the United States National Library of Medicine (NLM), a branch of the National Institutes of Health (NIH), served as a crucial platform, providing access to a vast array of biological databases and tools through its Rentrez R libraries. NCBI is a comprehensive resource that houses genetic, genomic, and biomedical data, including sequence data, literature references, and bioinformatics tools. This resource enabled us to retrieve relevant gene–disease associations and other critical information needed to explore the molecular underpinnings of periodontitis and its comorbidities.

Another important resource used in this study was the Online Mendelian Inheritance in Man (OMIM) database, a continuously updated catalog of human genes and genetic disorders, produced and curated at the Johns Hopkins School of Medicine (JHUSOM) Maryland, USA. OMIM provides detailed information on the relationships between genes and inherited diseases, which was instrumental in understanding the genetic basis of the diseases associated with periodontitis. By leveraging the information from OMIM, we were able to identify key genes linked to both periodontal disease and various systemic conditions, thereby contributing to the construction of the molecular comorbidity network.

Additionally, the DisGeNET database—maintained by the Integrative Biomedical Informatics (IBI) Group at Barcelona, Spain—played a pivotal role in our research. DisGeNET is a comprehensive platform that integrates data on gene–disease associations from various sources, including the scientific literature, curated databases, and genome-wide association studies (GWAS). This database was particularly valuable in identifying genes and proteins that are shared between periodontal disease and other systemic conditions. By using DisGeNET, we were able to systematically compile and analyze gene–disease interactions, facilitating the discovery of potential molecular pathways that link periodontitis with other chronic diseases. These resources, combined with robust bioinformatics tools, allowed us to uncover the complex molecular landscape of periodontal disease and its systemic implications. We will now proceed to introduce each of these tools.

### 4.1. Molecular Association Analysis

The primary genetic associations with periodontal disease were performed using the R Entrez and ROmim libraries. The R Entrez library (https://cran.r-project.org/web/packages/rentrez/index.html, accessed on 20 June 2024), provides a powerful interface for accessing biological data from the National Center for Biotechnology Information (NCBI) through the Entrez Programming Utilities (E-utilities) API. The ROmim library (https://github.com/davetang/romim, accessed on 20 June 2024), in turn was used to query the Online Mendelian Inheritance in Man (OMIM) database (https://www.omim.org/, accessed on 20 June 2024). Both database tools were used first to look up molecular information about periodontal disease; this information allowed to generate customized queries used to gather further information about diseases sharing genes and other biomolecules, as well as functional biological and biochemical pathways with periodontitis.

### 4.2. Molecular Comorbidity and Diseasome Networks

Once preliminary queries were carried out, a systematic molecular comorbidity network analysis and a diseasome network construction were performed by using the disgenet2r package (https://www.disgenet.org/disgenet2r, accessed on 20 June 2024). disgenet2r is an Application Programming Interface (API) to the DisGeNET database (https://www.disgenet.org/, accessed on 20 June 2024). DisGeNET is a discovery platform containing one of the largest publicly available collections of genes and variants associated to human diseases. DisGeNET consolidates information sourced from meticulously curated repositories, GWAS catalogs, animal models, and scholarly publications. The data within DisGeNET are uniformly annotated using standardized vocabularies and collaborative ontologies. Furthermore, DisGeNET offers various novel metrics aimed at aiding the prioritization of relationships between genotypes and phenotypes. An annotated pipeline to reproduce the workflow used to build comorbidity and diseasome networks for periodontitis can be found at https://github.com/CSB-IG/Periodontitis_diseasome (accessed on 20 June 2024).

#### 4.2.1. Molecular Comorbidity Network Reconstruction

A molecular comorbidity network, defined as a set of diseases that share associated genes, proteins, or biomolecular pathways with a given disease was built for periodontitis using the module of disgenet2r. Parameters were set to include only curated molecular interactions, i.e., genetic and biomolecular associations supported by direct experimental evidence. The Breadsearch score was set to 0.4 (medium-high) and the confidence score was set to 1 (maximum confidence based on the available experimental evidence).

Setting a proper threshold for the Breadsearch score and the confidence score in DisGeNET is crucial for ensuring the relevance and reliability of the gene-disease associations identified. The Breadsearch score measures the likelihood that a particular gene–disease association reported in the literature is accurate, based on text-mining approaches. A higher Breadsearch score indicates stronger evidence supporting the association. When setting a threshold for this score, it is common to choose a value that balances specificity and sensitivity, often in the range of 0.2 to 0.3. However, for more stringent analyses where high confidence in the associations is required, a threshold closer to 0.4 (as we did here) or higher may be more appropriate. This helps to filter out less certain associations and focus on those that are more likely to be biologically significant.

Similarly, the confidence score in DisGeNET represents the level of confidence in the gene–disease association based on curated data and integrated evidence from multiple sources. A proper threshold for the confidence score should be set according to the research needs. For exploratory studies, a lower threshold might be used to capture a broader range of potential associations, typically around 0.3 to 0.5. For studies requiring high confidence in the findings, a higher threshold of 0.7 or above is recommended. Since we are performing a comprehensive, mostly hypothesis-free analysis (a scenario prone to false positive findings), we have decided to set the highest possible score cut-off. This ensures that only associations with robust support from multiple data sources are considered, reducing the likelihood of false positives.

Molecular findings will be prioritized using the PLI scoring metric. PLI scores are a powerful tool for prioritizing genes in research, particularly when studying genetic diseases or conditions where mutations play a crucial role. Genes with high PLI scores are considered intolerant to loss-of-function (LoF) mutations, meaning that these mutations are less likely to be found in healthy populations because they are often deleterious or lethal. As a result, these genes are essential for normal cellular functions and development, making them prime candidates for further investigation, especially in the context of diseases where gene dysfunction is implicated.

High PLI scores also suggest that mutations in these genes are more likely to result in severe phenotypic consequences. By focusing on these genes, researchers can prioritize those that are more likely to contribute to serious or life-threatening conditions. This is particularly important in studies aiming to uncover the genetic basis of severe diseases or syndromes. Additionally, genes with high PLI scores are often involved in critical biological pathways. Studying these genes can provide insights into the underlying mechanisms of disease, as disruptions in these pathways are more likely to lead to pathological conditions. Prioritizing these genes can thus help in identifying key molecular pathways that are central to disease development and progression.

Moreover, since genes with high PLI scores are likely to play crucial roles in maintaining normal physiological functions, they can be targeted for therapeutic interventions. Understanding the role of these genes in disease can guide the development of drugs that either correct or compensate for the loss-of-function in these essential genes. In clinical settings, high PLI scores can be used to prioritize genes for genetic screening in patients. These features constitute the rationale that we have followed in the present work to prioritize genes involved in comorbidity relationships using PLI scores.

PODNet was used not only to build the comorbidity network but also to generate a heatmap of shared molecules among the different diseases and a Protein class heatmap that displays the functional information of the different biomolecules for the different diseases in the comorbidity network.

#### 4.2.2. Diseasome Network Reconstruction

Diseasomes are network-based approaches that allow researchers to explore the complex relationships between diseases, identify common underlying mechanisms, and uncover potential targets for therapeutic intervention. They serve as valuable resources for studying disease etiology, understanding disease comorbidities, and developing strategies for precision medicine and personalized healthcare. Diseasome networks centered in periodontal disease were built using the disease2disease_by_gene module of disgenet2r. Only curated associations were retained and the maximum number of connected diseases was considered. A disease–disease network centered on periodontitis and a full diseasome of periodontitis-associated diseases were built. A bar plot visualization depicting the number and fraction of shared genes were also generated using the disease2disease_by_gene module.

#### 4.2.3. Significance of the Literature and Database Mining Strategy

In order to assess the robustness of our results, we have resorted to a stringent choice of statistical significance parameters as explained above. Since we have supported our data and literature mining analysis on the largest, best curated, and most comprehensive databases (OMIM, PubMed, DisGenet) alternative sources will be almost surely of lower confidence. One approach that can be taken to pursue individual associations at a more in-depth level would be to look up for the specific genetic variation and variant effect databases such as dbSNP and using tools such as variant effect predictors and genomic studies such as the 1000 genomes and similar efforts. This will constitute the basis of a large number of additional projects and it is thus out of the scope of this work.

### 4.3. Functional Enrichment Analysis

Functional enrichment analysis, was performed by using the gProfileR2 tool (https://cran.r-project.org/web/packages/gprofiler2/index.html, accessed on 20 June 2024) to mine the g:Profiler webserver (https://biit.cs.ut.ee/gprofiler/gost, accessed on 20 June 2024). Hypergeometric tests were applied to the list of periodontitis comorbidity network genes. Only annotated genes were considered. Statistical significance was set for all categories with g:SCS-corrected *p*-value (equivalent to non-independent FDR tests, i.e., a form of multiple testing correction that considers that Gene Ontology (GO) categories, for instance, form a hierarchy and thus are not all independent) less than 0.05. This means that after correcting for multiple comparisons, the maximum allowable error is set to 5%. This is a common approach in high-throughput analytics in biomedical research. Values less than $1\times 10^{-16}$ were capped in the visualizations. gProfiler parameters used were as follows: version e111_eg58_p18_30541362, query date: 28 March 2024,14:37:37, organism: hsapiens.

A general workflow for the methods and analyses performed in this work is presented in Figure 8.

## 5. Conclusions

In conclusion, this study elucidates the molecular comorbidities of periodontal disease, providing valuable insights into the shared pathogenic mechanisms and potential therapeutic targets. By integrating literature mining and biological database interrogation, we have identified a broad spectrum of diseases sharing significant molecular overlap with periodontitis. This overlap encompasses critical genes, proteins, and biological pathways, highlighting potential shared pathogenic mechanisms. Network analysis revealed a highly interconnected web of diseases associated with periodontitis (its diseasome). This intricate network underscores the complex interplay between oral and systemic health, emphasizing the influence of periodontal disease on overall well-being. Furthermore, the identification of key hub genes within the network offers promising targets for therapeutic intervention strategies aimed at managing both periodontitis and its associated comorbidities. These findings pave the way for more holistic approaches to patient care, considering the multifaceted interactions between periodontal and systemic health. It is however relevant to notice that we have shown and discussed associations between periodontitis and various diseases based on shared genes. These studies alone are not enough to establish causality. Further research is needed to determine whether periodontitis directly contributes to the development of these diseases or if they share common risk factors or underlying mechanisms. Future investigations aimed at functionally validating the identified molecular connections and exploring targeted therapeutic strategies may thus hold significant promise for improving patient outcomes.

## Figures and Tables

**Figure 1 ijms-25-10161-f001:**
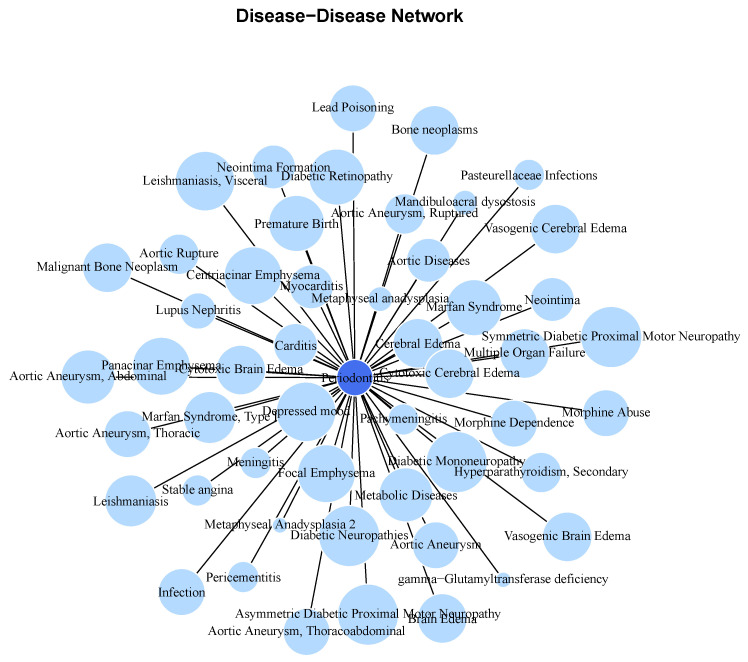
A periodontitis-centered molecular comorbidity network. Diseases sharing biomolecular players with periodontitis (dark blue node) are shown. Links indicate diseases sharing associated genes with periodontitis. Node sizes correspond to the Jaccard index (relative size of the intersection of the gene sets) with periodontitis.

**Figure 2 ijms-25-10161-f002:**
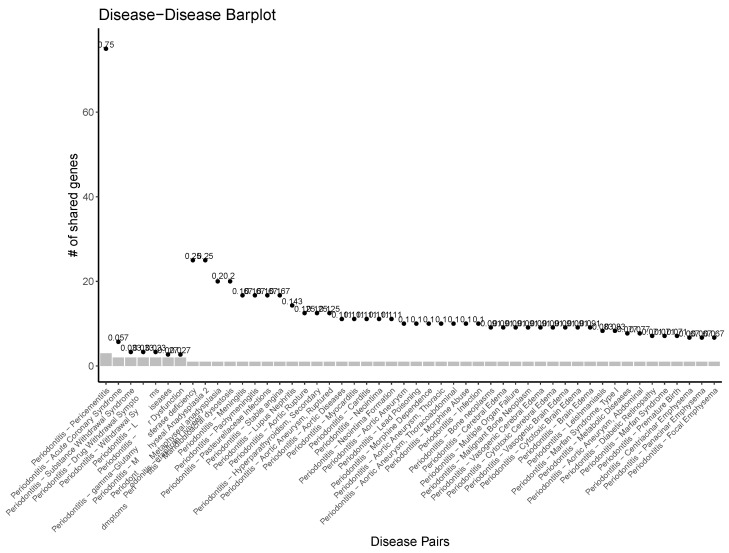
Numbers of genes shared between peridontitis and its main comorbidities. Diseases are also tagged according to the relative size of their intersection as given by Jaccard indices.

**Figure 3 ijms-25-10161-f003:**
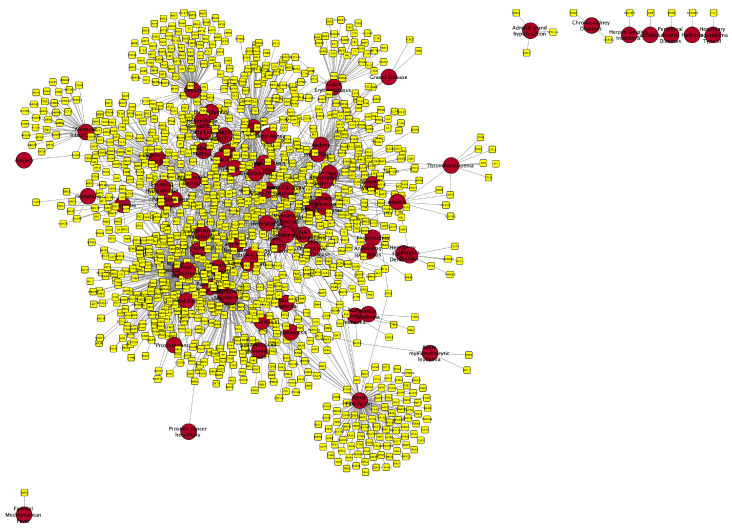
Molecular diseasome network depicting the interconnections created between periodontitis other diseases sharing common genetic associations. Every disease is connected to their associated genes. Red nodes correspond to the more significant health conditions associated with periodontitis. Yellow nodes correspond to the genes linked with each condition. This bipartite network consists of 80 disease nodes and 1197 gene nodes, with 2212 genetic associations between them. For visualization purposes, all disease nodes are depicted with the same size and all gene nodes are depicted with the same size irrespective of their degree of connectivity.

**Figure 4 ijms-25-10161-f004:**
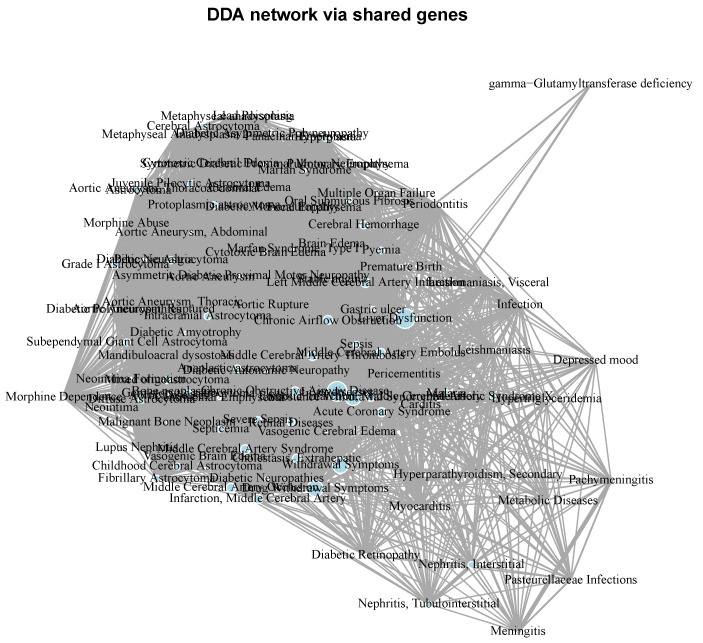
Diseasome network depicting the interconnections between periodontitis and its significant related conditions in a comprehensive manner. It considers molecular associations between any pair of conditions with a comorbidity relationship with periodontitis.

**Figure 5 ijms-25-10161-f005:**
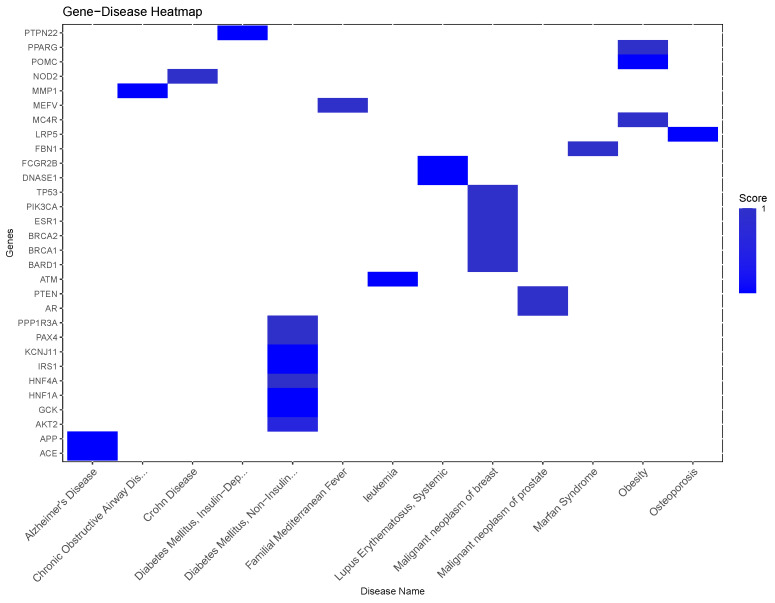
Top significant genes in the gene–disease map projected from the periodontitis-associated diseasome.

**Figure 6 ijms-25-10161-f006:**
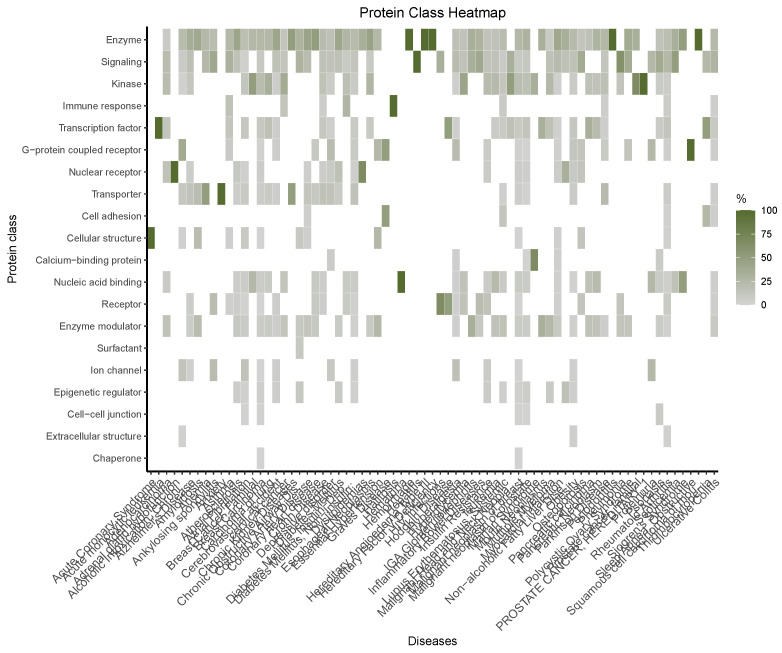
Heatmap of protein classes for the genes in the periodontitis-associated diseasome.

**Figure 7 ijms-25-10161-f007:**
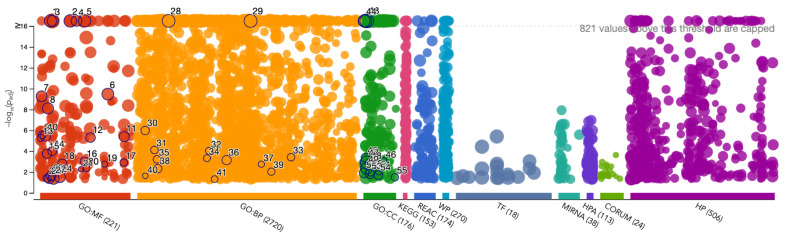
Functional enrichment analysis (over-representation) of genes shared by periodontitis and its comorbidity-related diseases. Notice that g:SCS-corrected *p*-values less than $1\times 10^{-16}$ are capped.

**Figure 8 ijms-25-10161-f008:**
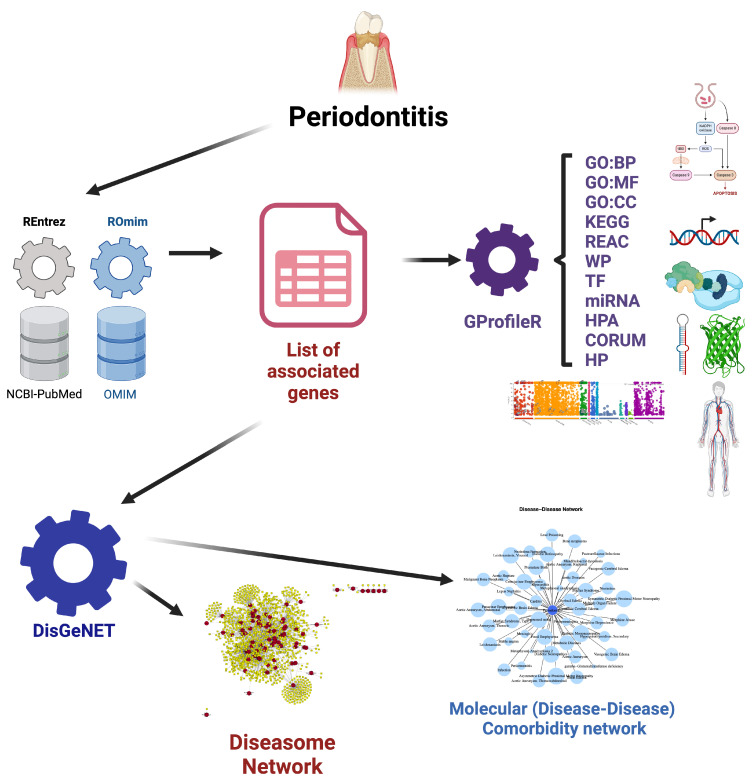
Workflow for the analysis presented. A list of periodontitis-associated genes was curated from a literature (NCBI-Pubmed) and database (OMIM) rigurous search. This list was used to mine high-confidence reported associations for diseases and conditions sharing genetic and molecular hits with periodontitis (DISGENET) to build comorbidity and diseasome Networks. The same list was also used to mine (GProfileR) biomolecular, pathway, and phenotype databases and to perform statistical tests for functional over-representation.

**Table 1 ijms-25-10161-t001:** Hub genes and their main associated diseases.

Hub Gene	Connectivity Degree	Top Diseases
TNF	24	Rheumatoid arthritis, ulcerative colitis, diabetes mellitus, myocardial infarction
IL6	23	Crohn disease, adenocarcinoma, malignant neoplasm of breast, atherosclerosis
PTGS2	14	Depressive disorder, esophageal neoplasms, cerebrovascular accident, coronary artery disease
IL10	13	Lupus erythematosus systemic, myocardial infarction, rheumatoid arthritis, ulcerative colitis
NOS3	12	Alzheimer’s disease, diabetes mellitus, coronary artery disease, malignant neoplasm of breast
IL1B	12	Hyperalgesia, asthma, myocardial infarction, rheumatoid arthritis
VEGFA	11	Psoriasis, adenocarcinoma, Alzheimer’s disease, atherosclerosis
BCL2	11	Polycystic ovary syndrome, diabetes mellitus, Alzheimer’s disease, malignant neoplasm of breast
STAT3	11	Psoriasis, ulcerative colitis, adenocarcinoma, atherosclerosis, malignant neoplasm of breast
LEP	11	Obesity, diabetes mellitus, depressive disorder, non-alcoholic fatty liver disease
TP53	11	Breast carcinoma, pancreatic carcinoma, mouth neoplasms, carcinoma of lung

**Table 2 ijms-25-10161-t002:** Selected molecular pathways showing statistically significant enrichment in the periodontitis molecular comorbidity network.

Source	Enriched Term	Code	Adjusted_*p*_Value
GO:MF	Signaling receptor binding	GO:0005102	1.6940040397 $\times \;10^{-72}$
GO:MF	Transcription factor binding	GO:0008134	6.414140091 $\times \;10^{-34}$
GO:BP	Humoral immune response mediated by circ. immunoglobulin	GO:0002455	0.00000105040
GO:BP	Mitotic recombination	GO:0006312	0.00007669851
KEGG	PI3K-Akt signaling pathway	KEGG:04151	2.499296336 $\times \;10^{-25}$
KEGG	AGE-RAGE signaling pathway in diabetic complications	KEGG:04933	5.788773392 $\times \;10^{-23}$
KEGG	FoxO signaling pathway	KEGG:04068	2.822200569 $\times \;10^{-16}$
KEGG	EGFR tyrosine kinase inhibitor resistance	KEGG:01521	5.951300224 $\times \;10^{-16}$
KEGG	JAK-STAT signaling pathway	KEGG:04630	9.66726614 $\times \;10^{-16}$
KEGG	HIF-1 signaling pathway	KEGG:04066	9.693733252 $\times \;10^{-15}$
KEGG	Th17 cell differentiation	KEGG:04659	1.09572282 $\times \;10^{-14}$
REAC	Interleukin-4 and Interleukin-13 signaling	HSA-6785807	2.492468333 $\times \;10^{-36}$
REAC	Cytokine Signaling in Immune system	HSA-1280215	6.751044139 $\times \;10^{-29}$
REAC	Signaling by GPCR	HSA-372790	2.413621576 $\times \;10^{-10}$
REAC	Regulation of IGF transport and uptake	HSA-381426	2.49720034 $\times \;10^{-9}$
WP	Focal adhesion PI3K Akt mTOR signaling pathway	WP:WP3932	1.756641872 $\times \;10^{-19}$
WP	A network map of Macrophage stimulating protein MSP signaling	WP:WP5353	1.202772226 $\times \;10^{-17}$
WP	TGF beta signaling pathway	WP:WP366	0.00000176565
WP	Fibrin complement receptor 3 signaling pathway	WP:WP4136	0.00000183517
WP	Prolactin signaling pathway	WP:WP2037	0.00000312440

## Data Availability

All the data generated in this study can be found in the Appendix A. Programming code to generate it can be found at the following public access repository: https://github.com/CSB-IG/Periodontitis_diseasome.

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
