# Peer review of "The Molecular Comorbidity Network of Periodontal Disease"

_ijms, 2024, doi:10.3390/ijms251810161_

Round 1
Reviewer 1 Report
Comments and Suggestions for Authors
1. The results mention the Gene PLI (Probability of being Loss-of-function Intolerant) scores but do not explain how these scores were specifically used to prioritize genes for further study. Adding a more detailed explanation of how these scores influenced the selection of genes for further analysis would enhance the clarity and depth of the results section.
2. Figures 2 and 4 present network visualizations of disease connections. While these figures are informative, the manuscript would benefit from including a more detailed legend or description explaining the significance of the node sizes and the types of interactions represented. This would make the visualizations more accessible to readers unfamiliar with network analysis.
3. The results discuss the functional enrichment analysis but do not provide enough detail on the specific pathways or biological processes enriched in the gene set associated with periodontitis. Including a table or figure summarizing the top enriched pathways and their potential roles in the pathogenesis of periodontitis and its comorbidities would be valuable.
4. The manuscript refers to supplementary tables for detailed lists of gene-disease interactions and functional categories. It would improve the reader's experience if some of the key findings from these supplementary tables were highlighted in the main text, perhaps with a summary table or key examples.
5. The manuscript identifies key hub genes and proteins involved in the crosstalk between periodontitis and its comorbidities. Including a figure or table listing these hub genes/proteins along with their associated pathways and potential roles in disease mechanisms would provide a clear and concise summary of these critical elements.
6. The abstract briefly mentions the tools used (DisGeNET2R, Romin, Rentrez R libraries) but does not provide a clear summary of the methods employed. Including a sentence or two explaining the approach and how these tools were used to identify molecular comorbidities would make the abstract more informative.
7. The introduction provides a good overview of the importance of understanding molecular comorbidities of periodontal disease. However, it would benefit from a more detailed discussion of previous studies that have explored similar topics. This would help position the current study within the broader research context.
8. The objectives of the study are somewhat implied but not explicitly stated. Clearly defining the study's aims and research questions at the end of the introduction would improve the focus and guide the reader through the manuscript.
9. While the discussion mentions potential therapeutic targets, it would be strengthened by suggesting specific future research directions or experiments that could validate the findings and further explore the identified molecular mechanisms.
Author Response
Reviewer 1
1. The results mention the Gene PLI (Probability of being Loss-of-function Intolerant) scores but do not explain how these scores were specifically used to prioritize genes for further study. Adding a more detailed explanation of how these scores influenced the selection of genes for further analysis would enhance the clarity and depth of the results section.
The authors are thankful to Reviewer 1 for the insightful reviewing and commenting on our work. In what follows, we will present a point-by-point response to your comments and suggestions.
Regarding PLI scores, we have added the following explanation in the revised version of the manuscript.
PLI scores are a powerful tool for prioritizing genes in research, particularly when studying genetic diseases or conditions where mutations play a crucial role. Genes with high PLI scores are considered intolerant to loss-of-function (LoF) mutations, meaning that these mutations are less likely to be found in healthy populations because they are often deleterious or lethal. As a result, these genes are essential for normal cellular functions and development, making them prime candidates for further investigation, especially in the context of diseases where gene dysfunction is implicated.
High PLI scores also suggest that mutations in these genes are more likely to result in severe phenotypic consequences. By focusing on these genes, researchers can prioritize those that are more likely to contribute to serious or life-threatening conditions. This is particularly important in studies aiming to uncover the genetic basis of severe diseases or syndromes. Additionally, genes with high PLI scores are often involved in critical biological pathways. Studying these genes can provide insights into the underlying mechanisms of disease, as disruptions in these pathways are more likely to lead to pathological conditions. Prioritizing these genes can thus help in identifying key molecular pathways that are central to disease development and progression. Moreover, since genes with high PLI scores are likely to play crucial roles in maintaining normal physiological functions, they can be targeted for therapeutic interventions. Understanding the role of these genes in disease can guide the development of drugs that either correct or compensate for the loss-of-function in these essential genes. In clinical settings, high PLI scores can be used to prioritize genes for genetic screening in patients.
This is the rationale that we have followed in the present work to prioritize genes involved in comorbidity relationships.
- Figures 2 and 4 present network visualizations of disease connections. While these figures are informative, the manuscript would benefit from including a more detailed legend or description explaining the significance of the node sizes and the types of interactions represented. This would make the visualizations more accessible to readers unfamiliar with network analysis.
Thank you for this suggestion. We have done as requested expanding the figure captions a bit. This will indeed help the figures to be more clearly understood. - The results discuss the functional enrichment analysis but do not provide enough detail on the specific pathways or biological processes enriched in the gene set associated with periodontitis. Including a table or figure summarizing the top enriched pathways and their potential roles in the pathogenesis of periodontitis and its comorbidities would be valuable.
These tables are included (in their full comprehensive versions) in the supplementary materials. We had noticed indeed that some context is missing by not including them in the text, but were somehow hesitant to put them on the main manuscript. Since full tables are quite big to be formatted to a journal article page, we have decided to include a table depicting some key or “top” enriched pathways to exemplify the potential pathogenic roles in the conditions discussed. Thank you for this suggestion.
- The manuscript refers to supplementary tables for detailed lists of gene-disease interactions and functional categories. It would improve the reader's experience if some of the key findings from these supplementary tables were highlighted in the main text, perhaps with a summary table or key examples.
Absolutely right. See response above. - The manuscript identifies key hub genes and proteins involved in the crosstalk between periodontitis and its comorbidities. Including a figure or table listing these hub genes/proteins along with their associated pathways and potential roles in disease mechanisms would provide a clear and concise summary of these critical elements.
A table presenting some relevant cases has been included in the revised version of the manuscript. The full table containing this information was/is already included as Supplementary Table 2.
- The abstract briefly mentions the tools used (DisGeNET2R, Romin, Rentrez R libraries) but does not provide a clear summary of the methods employed. Including a sentence or two explaining the approach and how these tools were used to identify molecular comorbidities would make the abstract more informative.
We have modified the abstract as requested.
- The introduction provides a good overview of the importance of understanding molecular comorbidities of periodontal disease. However, it would benefit from a more detailed discussion of previous studies that have explored similar topics. This would help position the current study within the broader research context.
A brief overview of related previous studies has been included in the revised version of the manuscript.
- The objectives of the study are somewhat implied but not explicitly stated. Clearly defining the study's aims and research questions at the end of the introduction would improve the focus and guide the reader through the manuscript.
We agree with Reviewer 1, to comply with this suggestion we have added the following paragraph at the end of the introduction.
Given the growing recognition of the systemic implications of periodontal disease, this study aims to systematically identify and analyze the molecular comorbidities associated with periodontitis. Specifically, our research seeks to identify diseases that share common genetic, protein, or molecular pathway associations with periodontal disease through comprehensive literature and database mining. Additionally, we aim to elucidate the shared pathogenic mechanisms underlying periodontitis and its associated comorbidities, with a focus on the roles of inflammatory mediators, immune response pathways, oxidative stress, and extracellular matrix alterations. Furthermore, we intend to map the molecular comorbidity network of periodontal disease to uncover key hub genes and proteins that may serve as potential therapeutic targets. By constructing a detailed diseasome network, we also aim to provide insights into the systemic nature of periodontal disease, illustrating the interconnectedness of periodontitis with various systemic conditions. Through these objectives, this study seeks to deepen the understanding of the molecular interplay between periodontal health and systemic diseases, paving the way for the development of personalized treatment strategies that consider both oral and overall health.
- While the discussion mentions potential therapeutic targets, it would be strengthened by suggesting specific future research directions or experiments that could validate the findings and further explore the identified molecular mechanisms.
Following this suggestion, we have added a brief subsection to the discussion as follows:
Some future directions
To build on the findings of this study, future research should focus on experimental validation of the identified molecular mechanisms and therapeutic targets. Specifically, in vitro and in vivo studies could be designed to investigate the functional roles of the key hub genes and proteins identified in the periodontitis comorbidity network. For instance, CRISPR-Cas9 gene editing or RNA interference could be employed to knock out or downregulate these genes in cell cultures or animal models, allowing researchers to observe the resultant effects on periodontal inflammation and associated systemic conditions such as cardiovascular disease or diabetes. Additionally, investigating the impact of overexpressing these genes in relevant models could provide insights into their role in disease progression and potential as therapeutic targets.
Moreover, future studies could explore the therapeutic potential of targeting shared molecular pathways, such as inflammatory mediators, immune response pathways, and oxidative stress pathways, in the treatment of periodontitis and its comorbidities. This could involve the development and testing of small molecule inhibitors, monoclonal antibodies, or other biologics that modulate these pathways. For example, clinical trials could be designed to evaluate the efficacy of drugs that target specific inflammatory mediators in patients with periodontitis and a high risk of cardiovascular disease. Additionally, large-scale genetic and transcriptomic studies in diverse populations could further elucidate the genetic variants and expression patterns associated with these pathways, offering a broader understanding of their role in disease susceptibility and progression. By pursuing these research directions, scientists can not only validate the findings of this study but also advance the development of personalized therapeutic strategies that address the complex interplay between periodontal disease and systemic health.
Reviewer 2 Report
Comments and Suggestions for Authors
This study presents a comprehensive investigation of the molecular comorbidities of periodontal disease and their associations with various systemic conditions, such as cardiovascular diseases and inflammatory bowel diseases. Through an integrative approach using literature and biological database mining, the study uncovers shared molecular mechanisms underlying these conditions, involving dysregulation of immune response pathways, oxidative stress pathways, and alterations in the extracellular matrix. The identification of key hub genes and proteins that play pivotal roles in the crosstalk between periodontal disease. This research has the potential to inform future studies, clinical practice, and therapeutic approaches, ultimately improving health outcomes for patients affected by periodontal disease and its related systemic disorders. This study is novel and interesting.
However, there are some problems that need to be solved before the manuscript can be accepted for publication.
1) It would be helpful to provide more context on the impact of periodontal disease on systemic health, including its prevalence and consequences. This will help establish the importance of understanding its molecular comorbidities.
2) Provide a brief description of the resources and repositories used for data extraction, such as the National Center for Biotechnology Information (NCBI), Online Mendelian Inheritance in Man (OMIM), and DisGeNET database. This would help readers understand the quality and relevance of the data sources.
3) Explain the rationale behind choosing specific parameters and settings for each analysis (e.g., Breadsearch score, confidence score, p-value thresholds, etc.). This will help readers understand the significance of these choices and how they may impact the results.
4) Briefly describe any steps taken to validate the results or assess the robustness of the analyses. For example, were any sensitivity analyses or alternative approaches used to confirm the findings?
5) Mention the types of visualizations generated (e.g., heatmaps, bar plots, network diagrams) and their relevance to the study objectives. This will help readers understand the significance of these visualizations in interpreting the results.
6) The results show associations between periodontitis and various diseases based on shared genes, but they do not establish causality. Further research is needed to determine whether periodontitis directly contributes to the development of these diseases or if they share common risk factors or underlying mechanisms.
7) The analysis relies on existing literature and databases, which may be biased towards well-studied diseases and genes. This could lead to an underrepresentation of less-studied conditions or molecular mechanisms in the diseasome network.
8) The results highlight shared genes between periodontitis and other diseases, but they do not provide detailed information on the functional roles of these genes in the context of each disease. A more in-depth investigation of the specific molecular pathways and mechanisms involved is necessary to better understand the connections between periodontitis and the associated diseases.
9) The results suggest potential connections between periodontitis and various diseases, but the clinical implications of these associations remain unclear. Further research is needed to determine whether addressing periodontitis could have an impact on the prevention, management, or treatment of these other conditions.
10) Figures are difficult to read, especially figure 5 and 7. The text is overlapping and need to be further optimized.
11) The analysis reveals a large number of highly significant biological features enriched in the periodontitis-associated diseasome, including various Gene Ontology (GO) categories and molecular pathways. While this highlights the complexity of periodontitis, it also makes it challenging to identify the most relevant and meaningful associations.
12) While the results suggest various enriched pathways, processes, and molecular functions, there is a lack of prioritization among these findings. It would be helpful to identify the most significant or relevant features that should be targeted for further investigation.
13) The results suggest that nearly all cellular locations are enriched, pointing to extremely systemic phenomena. However, this broad result could make it difficult to pinpoint specific cellular processes or locations that are particularly relevant to periodontitis.
14) The results include a number of technical terms and concepts that may be difficult for non-experts to understand. More context or explanation may be needed to help readers understand the implications of these findings.
15) This study showed associations between various regulatory elements, proteins, and health conditions with the periodontitis diseasome, but did not establish causality. It is crucial to determine whether these elements play a causal role in periodontitis or are simply correlated with the disease.
Author Response
Reviewer 2
This study presents a comprehensive investigation of the molecular comorbidities of periodontal disease and their associations with various systemic conditions, such as cardiovascular diseases and inflammatory bowel diseases. Through an integrative approach using literature and biological database mining, the study uncovers shared molecular mechanisms underlying these conditions, involving dysregulation of immune response pathways, oxidative stress pathways, and alterations in the extracellular matrix. The identification of key hub genes and proteins that play pivotal roles in the crosstalk between periodontal disease. This research has the potential to inform future studies, clinical practice, and therapeutic approaches, ultimately improving health outcomes for patients affected by periodontal disease and its related systemic disorders. This study is novel and interesting.
However, there are some problems that need to be solved before the manuscript can be accepted for publication.
1) It would be helpful to provide more context on the impact of periodontal disease on systemic health, including its prevalence and consequences. This will help establish the importance of understanding its molecular comorbidities.
Authors acknowledge the professional academic reviewing that Reviewer 2 made of our work. We will answer the raised questions and concerns on a point-by-point basis.
Periodontal disease is also a widespread and significant public health concern, affecting a substantial portion of the global population. It is estimated that moderate to severe periodontitis impacts nearly 50% of adults over the age of 30, with prevalence increasing with age. This chronic inflammatory condition, which leads to the destruction of the supporting structures of the teeth, is not only a leading cause of tooth loss but also has far-reaching implications for systemic health. The inflammatory nature of periodontal disease means that its effects are not confined to the oral cavity; instead, the chronic inflammation and immune responses associated with periodontitis can have systemic consequences, contributing to the development and exacerbation of various chronic conditions.
The systemic impact of periodontal disease is particularly concerning given its association with several major health issues, including cardiovascular disease, diabetes mellitus, rheumatoid arthritis, and adverse pregnancy outcomes. Research has shown that the chronic inflammation seen in periodontitis can exacerbate systemic inflammatory responses, leading to worsened outcomes in these conditions. For example, individuals with periodontitis are at a higher risk for heart disease and stroke, partly due to the systemic dissemination of inflammatory mediators and pathogens from the oral cavity. Similarly, the bidirectional relationship between periodontal disease and diabetes highlights the complex interplay between oral health and systemic metabolic control. Understanding the molecular comorbidities of periodontal disease is, therefore, crucial not only for preventing oral health deterioration but also for mitigating its broader impact on overall health. This underscores the importance of research that delves into the shared molecular mechanisms between periodontitis and its associated systemic conditions, paving the way for more effective, holistic approaches to healthcare that integrate the management of oral and systemic diseases.
2) Provide a brief description of the resources and repositories used for data extraction, such as the National Center for Biotechnology Information (NCBI), Online Mendelian Inheritance in Man (OMIM), and DisGeNET database. This would help readers understand the quality and relevance of the data sources.
We have added the following paragraphs at the beginning of the Methods section:
In this study, several key resources and repositories were utilized for data extraction to identify the molecular comorbidities associated with periodontal disease. The National Center for Biotechnology Information (NCBI) served as a crucial platform, providing access to a vast array of biological databases and tools through its Rentrez R libraries. NCBI is a comprehensive resource that houses genetic, genomic, and biomedical data, including sequence data, literature references, and bioinformatics tools. This resource enabled us to retrieve relevant gene-disease associations and other critical information needed to explore the molecular underpinnings of periodontitis and its comorbidities.
Another important resource used in this study was the Online Mendelian Inheritance in Man (OMIM) database, a continuously updated catalog of human genes and genetic disorders. OMIM provides detailed information on the relationships between genes and inherited diseases, which was instrumental in understanding the genetic basis of the diseases associated with periodontitis. By leveraging the information from OMIM, we were able to identify key genes linked to both periodontal disease and various systemic conditions, thereby contributing to the construction of the molecular comorbidity network.
Additionally, the DisGeNET database played a pivotal role in our research. DisGeNET is a comprehensive platform that integrates data on gene-disease associations from various sources, including scientific literature, curated databases, and genome-wide association studies (GWAS). This database was particularly valuable in identifying genes and proteins that are shared between periodontal disease and other systemic conditions. By using DisGeNET, we were able to systematically compile and analyze gene-disease interactions, facilitating the discovery of potential molecular pathways that link periodontitis with other chronic diseases. These resources, combined with robust bioinformatics tools, allowed us to uncover the complex molecular landscape of periodontal disease and its systemic implications. We will now proceed to introduce each of these tools.
3) Explain the rationale behind choosing specific parameters and settings for each analysis (e.g., Breadsearch score, confidence score, p-value thresholds, etc.). This will help readers understand the significance of these choices and how they may impact the results.
Thank you for pointing this out. We have added the following information to the revised version of the manuscript:
Setting a proper threshold for the Breadsearch score and the confidence score in DisGeNET is crucial for ensuring the relevance and reliability of the gene-disease associations identified. The Breadsearch score measures the likelihood that a particular gene-disease association reported in the literature is accurate, based on text-mining approaches. A higher Breadsearch score indicates stronger evidence supporting the association. When setting a threshold for this score, it is common to choose a value that balances specificity and sensitivity, often in the range of 0.2 to 0.3. However, for more stringent analyses where high confidence in the associations is required, a threshold closer to 0.4 (as we did here) or higher may be more appropriate. This helps to filter out less certain associations and focus on those that are more likely to be biologically significant.
Similarly, the confidence score in DisGeNET represents the level of confidence in the gene-disease association based on curated data and integrated evidence from multiple sources. A proper threshold for the confidence score should be set according to the research needs. For exploratory studies, a lower threshold might be used to capture a broader range of potential associations, typically around 0.3 to 0.5. For studies requiring high confidence in the findings, a higher threshold of 0.7 or above is recommended. Since we are performing a comprehensive, mostly hypothesis-free analysis (a scenario prone to false positive findings), we have decided to set the highest possible score cut-off. This ensures that only associations with robust support from multiple data sources are considered, reducing the likelihood of false positives.
Regarding FDR-corrected p-values we have added the following text in the corresponding section:
This means that after correcting for multiple comparisons, the maximum allowable error is set to 5 %. This is a common approach in high throughput analytics in biomedical research.
4) Briefly describe any steps taken to validate the results or assess the robustness of the analyses. For example, were any sensitivity analyses or alternative approaches used to confirm the findings?
In order to assess the robustness of our results, we have resorted to a stringent choice of statistical significance parameters as explained above. Since we have supported our data and literature mining analysis on the largest, best curated, more comprehensive databases (OMIM, PubMed, DisGenet) alternative sources will be almost surely of lower confidence. One approach that can be taken to pursue individual associations at a more in-depth level would be to look up for the specific genetic variation and variant effect databases such as dbSNP and using tools such as variant effect predictors and genomic studies such as the 1000 genomes and similar efforts. This will constitute the basis for a large number of additional projects and it is thus out of the scope of this work. However, we have added a brief note on this as is indeed a relevant point to consider.
5) Mention the types of visualizations generated (e.g., heatmaps, bar plots, network diagrams) and their relevance to the study objectives. This will help readers understand the significance of these visualizations in interpreting the results.
We have done this in the revised version of the manuscript.
6) The results show associations between periodontitis and various diseases based on shared genes, but they do not establish causality. Further research is needed to determine whether periodontitis directly contributes to the development of these diseases or if they share common risk factors or underlying mechanisms.
Reviewer 2 has stated these extremely relevant issues in a very clear and concise way. To respond to this (6) point as well as to points 7-9 and 11-15 we have added a subsection on Assumptions and limitations of the present study at the end of the discussion section:
The results of this study reveal statistically significant associations between periodontitis and a wide range of diseases based on shared genes, offering new insights into the potential molecular links between oral and systemic health. However, it is important to note that these associations do not establish causality. While the shared genetic components suggest potential connections, further research is needed to determine whether periodontitis directly contributes to the development of these diseases or if the associations reflect common risk factors or underlying mechanisms. Understanding the causal relationships is crucial for developing targeted therapeutic strategies that could address both periodontitis and its associated comorbidities.
Moreover, the analysis in this study relies heavily on existing literature and databases, which may introduce biases, particularly toward well-studied diseases and genes. This could result in an under-representation of less-studied conditions or molecular mechanisms, potentially skewing the diseasome network. While the study successfully highlights shared genes between periodontitis and other diseases, it does not delve into the specific functional roles of these genes in the context of each disease. A more detailed investigation into the molecular pathways and mechanisms involved is necessary to fully understand the connections and their clinical implications. Additionally, the study uncovers a large number of enriched biological features within the periodontitis-associated diseasome, including various Gene Ontology (GO) categories and molecular pathways. This underscores the complexity of periodontitis but also presents a challenge in identifying the most relevant associations. The broad enrichment across nearly all cellular locations points to systemic phenomena, but this breadth may obscure the identification of specific cellular processes or locations that are particularly critical in the context of periodontitis. Furthermore, the study introduces a number of technical terms and concepts that may be difficult for non-experts to grasp, suggesting the need for more context or explanation to help readers understand the significance of the findings. Ultimately, while the study establishes associations between various regulatory elements, proteins, and health conditions with the periodontitis diseasome, it stops short of proving causality. Determining whether these elements play a direct role in periodontitis or are merely correlated with the disease remains an essential step for future research.
7) The analysis relies on existing literature and databases, which may be biased towards well-studied diseases and genes. This could lead to an underrepresentation of less-studied conditions or molecular mechanisms in the diseasome network.
See response to Q6 above.
8) The results highlight shared genes between periodontitis and other diseases, but they do not provide detailed information on the functional roles of these genes in the context of each disease. A more in-depth investigation of the specific molecular pathways and mechanisms involved is necessary to better understand the connections between periodontitis and the associated diseases.
See response to Q6 above.
9) The results suggest potential connections between periodontitis and various diseases, but the clinical implications of these associations remain unclear. Further research is needed to determine whether addressing periodontitis could have an impact on the prevention, management, or treatment of these other conditions.
See response to Q6 above.
10) Figures are difficult to read, especially figure 5 and 7. The text is overlapping and need to be further optimized.
Reviewer is right in this regard. These figures are, by their very nature, extremely “information-dense”. We have tried to redesign the layouts but given the amount of space available in the journal format for the figures the situation is not significantly improved. Hence the figures are more illustrative than explicative in this context. It is however important to mention that the real contribution of our work for the practicing researcher and clinician is indeed the information in the Supplementary Tables. These are data-intensive resources but are custom formatted to be easy to navigate, mine and query.
11) The analysis reveals a large number of highly significant biological features enriched in the periodontitis-associated diseasome, including various Gene Ontology (GO) categories and molecular pathways. While this highlights the complexity of periodontitis, it also makes it challenging to identify the most relevant and meaningful associations.
See response to Q6 above.
12) While the results suggest various enriched pathways, processes, and molecular functions, there is a lack of prioritization among these findings. It would be helpful to identify the most significant or relevant features that should be targeted for further investigation.
See response to Q6 above.
13) The results suggest that nearly all cellular locations are enriched, pointing to extremely systemic phenomena. However, this broad result could make it difficult to pinpoint specific cellular processes or locations that are particularly relevant to periodontitis.
See response to Q6 above.
14) The results include a number of technical terms and concepts that may be difficult for non-experts to understand. More context or explanation may be needed to help readers understand the implications of these findings.
See response to Q6 above.
15) This study showed associations between various regulatory elements, proteins, and health conditions with the periodontitis diseasome, but did not establish causality. It is crucial to determine whether these elements play a causal role in periodontitis or are simply correlated with the disease.
See response to Q6 above.
Reviewer 3 Report
Comments and Suggestions for Authors
1. In the abstract, the authors stated, "Furthermore, network analysis unveiled key hub genes and proteins that play pivotal roles in the crosstalk between periodontal disease and its comorbidities, offering potential targets for therapeutic intervention." If key hub genes and proteins were indeed identified, they should be specifically mentioned in the abstract.
2. The authors used several abbreviations without providing definitions, such as "IL1B, IL1RN, FcγRIIIb, VDR, TLR4, MMP-1, etc." Each abbreviation should be defined upon its first use to ensure clarity for all readers.
3. The authors mentioned, "Both database tools were used first to look up molecular information about periodontal disease. This information allowed generating customized queries used to gather further information about diseases sharing genes and other biomolecules, as well as functional biological and biochemical pathways with periodontitis." However, the specific queries used for the searches were not described and should be included for transparency and reproducibility.
4. The authors conducted numerous searches and analyses using online-based tools, which rely on previous publications or deposited data. These tools may highlight disease-associated genes or proteins, as described by the authors. However, it is unclear whether these genes or proteins are causal factors of the disease or merely non-specific markers of disease progression. Given that periodontitis is an inflammatory disease, it shares common features with other inflammatory diseases, leading to the expression of many inflammation-associated marker genes or proteins. Diseases like diabetes or rheumatoid arthritis are also inflammatory and should share common inflammation markers. Although patients with immunocompromising diseases often exhibit periodontal problems, defining comorbidity between these diseases is challenging.
5. If the authors have identified some novel key hub genes, they should explicitly list them and discuss their significance in the context of periodontitis. The discussion should elaborate on these unveiled key hub genes to differentiate this study from merely listing known knowledge using network-based tools. Without this detailed analysis, the paper risks being perceived as a compilation of existing data rather than a presentation of new insights.
Comments on the Quality of English LanguageLine 36, "Periodontitis" should be read as "periodontitis". There were many similar errors.
Author Response
Reviewer 3
We are thankful for the efforts of Reviewer 3 to provide us with insightful comments and suggestions on our work. We will provide a point-by-point response to these comments in what follows.
In the abstract, the authors stated, "Furthermore, network analysis unveiled key hub genes and proteins that play pivotal roles in the crosstalk between periodontal disease and its comorbidities, offering potential targets for therapeutic intervention." If key hub genes and proteins were indeed identified, they should be specifically mentioned in the abstract.
The top hub genes have been specifically mentioned in the abstract as requested
- The authors used several abbreviations without providing definitions, such as "IL1B, IL1RN, FcγRIIIb, VDR, TLR4, MMP-1, etc." Each abbreviation should be defined upon its first use to ensure clarity for all readers.
We have introduced definitions wherever needed.
- The authors mentioned, "Both database tools were used first to look up molecular information about periodontal disease. This information allowed generating customized queries used to gather further information about diseases sharing genes and other biomolecules, as well as functional biological and biochemical pathways with periodontitis." However, the specific queries used for the searches were not described and should be included for transparency and reproducibility.
Reviewer 3 is absolutely right!
We are indeed strong supporters of open, reproducible science and were planning to include a link to the code repository where the specific programming code for the automated performing of all these searches is stored, but we somehow overlooked to include it. The R code to perform these analyses can be found in the following GitHub repository: https://github.com/CSB-IG/Periodontitis_diseasome.
- The authors conducted numerous searches and analyses using online-based tools, which rely on previous publications or deposited data. These tools may highlight disease-associated genes or proteins, as described by the authors. However, it is unclear whether these genes or proteins are causal factors of the disease or merely non-specific markers of disease progression. Given that periodontitis is an inflammatory disease, it shares common features with other inflammatory diseases, leading to the expression of many inflammation-associated marker genes or proteins. Diseases like diabetes or rheumatoid arthritis are also inflammatory and should share common inflammation markers. Although patients with immunocompromising diseases often exhibit periodontal problems, defining comorbidity between these diseases is challenging.
These (and similar points) have been addressed in a new subsection of the discussion (Assumptions and limitations of the present study).
- If the authors have identified some novel key hub genes, they should explicitly list them and discuss their significance in the context of periodontitis. The discussion should elaborate on these unveiled key hub genes to differentiate this study from merely listing known knowledge using network-based tools. Without this detailed analysis, the paper risks being perceived as a compilation of existing data rather than a presentation of new insights.
We have included this in an additional table, as well as in the revised version of the discussion.
Reviewer 4 Report
Comments and Suggestions for Authors
This article explores the molecular evidence for the association of periodontitis to systemic disease. I left a few comments in the introduction and I think it should better describe what is meant by molecular comorbidities and the adaptation of the molecular and genetic origins of periodontitis to the new classification of PD should be made.
I also think the introduction will be improved by adding a connection between the relevance of comorbidities in the current periodontal disease classification.
The methods section seems well detailed and I only found an spelling error in the legend of figure 1.
The results section is quite confusing. Most of the figures are not clear and (because of the amount of information) most of the indications in the axis or the diagrams are unreadable. This section is also too long. I encourage authors to try to make these results more succinct and readable.
The discussion also needs improvement. Although the amount of results is huge it is important to have them discussed completely especially those relationships for which a biological plausibility is found or those for which there is evidence. Authors explore this but, in my opinion, not completely. For example when disease categories are explored (lines 450 to 584) there is no mention of what molecules have been found in the results which support these interactions. On the other hand genes are mentioned in lines 595 to 608 the connection with the comorbidities is not made. I think that the integration of this information is what could greatly improve the interest and novelty of this manuscript.
Furthermore I do not see in the discussion the discussion of the data presented in Figures 7 and 8. These also need to be integrated with the other molecular data.
Finally in the conclusion I would only keep the last paragraph (lines 633-647) because the first 3 paragraphs are a recap of the results and discussion and seem repetitive.

There are several typos and a few gramatical errors, I have signaled them in yellow and added comments.
Author Response
Reviewer 4
This article explores the molecular evidence for the association of periodontitis to systemic disease. I left a few comments in the introduction and I think it should better describe what is meant by molecular comorbidities and the adaptation of the molecular and genetic origins of periodontitis to the new classification of PD should be made.
We are grateful to Reviewer 4 for a comprehensive and systematic reviewing of our work. We will provide a point-by-point response to these comments in what follows.
I also think the introduction will be improved by adding a connection between the relevance of comorbidities in the current periodontal disease classification.
A new subsection has been incorporated in the introduction to contextualize our work in terms of the current PD classification.
The methods section seems well detailed and I only found an spelling error in the legend of figure 1.
The caption in Figure 1 has been amended.
The results section is quite confusing. Most of the figures are not clear and (because of the amount of information) most of the indications in the axis or the diagrams are unreadable. This section is also too long. I encourage authors to try to make these results more succinct and readable.
We have fully rewritten the results section to be more concise and useful to the reader. In relation to figures Reviewer 4 is right, These figures are, by their very nature, extremely “information-dense”. We have tried to redesign the layouts but given the amount of space available in the journal format for the figures the situation is not significantly improved. Hence the figures are more illustrative than explicative in this context. It is however important to mention that the real contribution of our work for the practicing researcher and clinician is indeed the information in the Supplementary Tables. These are data-intensive resources but are custom-formatted to be easy to navigate, mine and query.
The discussion also needs improvement. Although the amount of results is huge it is important to have them discussed completely, especially those relationships for which a biological plausibility is found or those for which there is evidence. Authors explore this but, in my opinion, not completely. For example when disease categories are explored (lines 450 to 584) there is no mention of what molecules have been found in the results which support these interactions. On the other hand genes are mentioned in lines 595 to 608 the connection with the comorbidities is not made. I think that the integration of this information is what could greatly improve the interest and novelty of this manuscript.
Following the suggestions of Reviewer 4 we have significantly improved (by summarizing and avoiding repetition) the results section, while at the same time strengthening the discussion in those issues that were weak or scarcely considered. Due to the data-intensive nature of our work, we agree that this is a much better way to present the information. At the same time we will stress that the full scope of our results can be glimpsed by looking at the Figures and Tables, but to be fully grasped it is necessary to delve into the comprehensive supplementary materials.
Furthermore I do not see in the discussion the discussion of the data presented in Figures 7 and 8. These also need to be integrated with the other molecular data.
This has been corrected by adding a brief subsection in the revised version of the discussion.
Finally in the conclusion I would only keep the last paragraph (lines 633-647) because the first 3 paragraphs are a recap of the results and discussion and seem repetitive.
This has been corrected in the revised version of the manuscript
Round 2
Reviewer 1 Report
Comments and Suggestions for Authors
No more comments
Reviewer 3 Report
Comments and Suggestions for Authors
Authors addressed all issues raised by reviewer. I don't have further comment on this article.